

# Review of the genus *Microlaimus* de Man, 1880 with an illustrated guide for species identification

Andre M. Esteves, Alex Manoel and Patricia F. Neres

Department of Zoology, UFPE, Universidade Federal de Pernambuco, Recife, Pernambuco, Brazil

## ABSTRACT

Among the representatives of the family Microlaimidae, the genus *Microlaimus* is the most species richeness. However, the most recent literature on this genus presents species lists that diverge, in terms of composition and the number of species considered valid for *Microlaimus*. The morphological characteristics of this genus overlap with those of other genera of Microlaimidae, making the taxonomy of this genus complex. In the present study, we reviewed the species of the genus *Microlaimus*, as well as species included in genera of Microlaimidae that are morphologically similar to this taxon. Groups of species that share certain characteristics were created and represented in an illustrated guide for intraspecific identification of *Microlaimus*. The position of the amphidial fovea in relation to the anterior end of the body, provided important taxonomic information that was used to distinguish *Microlaimus* species. The presence or absence of cuticular ornamentation, the size of the spicules, the absence/presence of gubernaculum and the amphidial fovea position in relation to the anterior end of the body were characteristics used to separate the groups. Furthermore, the morphology of the male copulatory structures, as well as the composition of the cephalic arrangement (papillae, setiform papillae and setae) and the length of the cephalic setae in relation to head diameter, were also relevant for the characterization of the species.

## INTRODUCTION

The family Microlaimidae was erected by *Micoletzky (1922)* based on the genus *Microlaimus de Man, 1880*. Currently, this family comprises 13 genera (*Tchesunov, Jeong & Lee, 2021*), which include more than 150 species (*WoRMS Editorial Board, 2024*).

*Microlaimus* is the most diverse genus of the family, commonly found in the marine environment, but there are also records of only a small number of species in the freshwater environment (*Tchesunov, 2014*). The genus is widely distributed, with representatives in all oceans of the world, however a greater number of species have been recorded for the North Atlantic (*WoRMS Editorial Board, 2024*). The genus is present in a wide variety of habitat types, standing out in terms of abundance even in coral reef sediments (*Lin et al., 2025*; *Ng et al., 2022*), as well as in deep regions (*Miljutin & Miljutina, 2009*; *Schnier et al., 2023*). Regarding the species of this genus, the majority were described for continental shelf

Corresponding author
Andre M. Esteves,
andresteves.ufpe@gmail.com

environments, mainly on sandy beaches (*Lima, 2016*). Over time, several taxonomic studies have contributed significantly to the available knowledge on Microlaimidae (*Jensen, 1978*; *Lorenzen, 1994*; *Kovalyev & Tchesunov, 2005*; *Decraemer & Smol, 2006*; *Tchesunov, 2014*). *Decraemer & Smol (2006)* constructed a key to identify the 10 genera known and classified in Microlaimidae. Since then, three genera have been added to this family: *Maragnopsia Leduc, 2016*; *Macrodontium Armenteros, Vincx & Decraemer, 2010*; *Jejulaimus Tchesunov, Jeong & Lee, 2021*. In all these studies, comparative tables were presented that compile information about the genera of Microlaimidae, indicating the main differences between them. More recently, *Manoel, Neres & Esteves (2024a)* added new information to the diagnosis of *Ixonema Lorenzen, 1971* and highlighted the main taxonomic characteristics that segregate *Ixonema* from the genus *Bathynox* (*Bussau, 1993*; *Bussau & Vopel, 1999*).

The most recent literature presents lists that diverge in terms of composition and number of valid species for the genus *Microlaimus*, ranging from 86 to 93 species (*Tchesunov, Jeong & Lee, 2021*; *Lima, Neres & Esteves, 2022*; *Guo, Wang & Wang, 2023*; *WoRMS Editorial Board, 2024*; *Nemys, 2024*).

This can be observed in several reviews that record transfers involving species from genera that are morphologically similar to *Microlaimus* (such as *Aponema Jensen, 1978*, *Bolbolaimus Cobb, 1920* and *Calomicrolaimus Lorenzen, 1976*), including the transfer of species from genera that belong to a distinct family, as is the case of *Molgolaimus Ditlevsen, 1921* (Desmodoridae *De Coninck, 1965*) (*Jensen, 1978*; *Kovalyev & Miljutina, 2009*; *Miljutin & Miljutina, 2009*; *Tchesunov, 2014*; *Leduc, 2016*). This is a result of disagreements about which morphological characters should be used to establish differences between such genera (*Leduc, 2016*). *Jensen (1978)* mentioned the difficulty of constructing an identification key for the genus, indicating that this is not only a reflection of the close relationship between the species but is also associated with the number of inadequate descriptions. This set of factors makes the taxonomy of *Microlaimus* complex.

## RATIONALE FOR THIS REVIEW

The morphological characteristics of this genus overlap with those of other genera of Microlaimidae, which makes it difficult to differentiate it from other representatives of this family (*Platt & Warwick, 1988*; *Decraemer & Smol, 2006*). Consequently, the need to review the genera of this family has been previously mentioned in the literature (*Platt & Warwick, 1988*; *Decraemer & Smol, 2006*).

In this study, we present a tool that aims to facilitate the intraspecific identification of *Microlaimus*. Thus, we created an "illustrated guide" with the species considered here as valid, following review. The species were strategically grouped based on morphological characters that are easy to visualize and frequently present in the descriptions of the species of the genus (textual or through images). In addition, the species classified in genera that are morphologically close to *Microlaimus* were reanalyzed to confirm whether they were placed in the correct taxon. Finally, we highlight which morphological characters are most relevant for the intraspecific identification of the genus.

## AUDIENCE

This manuscript will be useful to all experts in marine nematode taxonomy throughout the world.

## METHODOLOGY

Initially, we performed a review of the *Microlaimus* species present in the *Nemys (2024)* and *WoRMS Editorial Board (2024)* database, in addition to those listed by *Lima, Neres & Esteves (2022)*. To choose which species of Microlaimidae genera would be evaluated, we consulted *Tchesunov (2014)* (section dedicated to the Superfamily Microlaimoidea *Micoletzky, 1922*) and the comparative table provided by *Tchesunov, Jeong & Lee (2021)*, where the morphological differences between the genera of Microlaimidae were analyzed. The genera that presented a relevant number of species transferred to *Microlaimus* (or the opposite) were considered morphologically close (*e.g.*, *Bolbolaimus*, *Aponema*). These genera were evaluated in order to verify whether these species are positioned in the correct genus.

Following this evaluation, the species considered valid and belonging to *Microlaimus* were separated into groups and, when relevant, into subgroups using easily identifiable morphological characteristics that are frequently present in the descriptions of species of this taxon. Plates with drawings of the anterior and posterior end of each species (principally of the holotype when available and alternatively of paratypes) were prepared. Within each group/subgroup formed, the species were arranged in alphabetical order. The separation presented does not reflect any phylogenetic relationships between the species.

Preliminarily, the groups presented in Table 1 were formed based on the following criteria: Group 1: spicules equal to or greater than 2.8 cloacal diameters (dc); Group 2: gubernaculum absent and Group 3: cuticular ornamentation present. The species that present short spicules (up to 2.3 dc), gubernaculum present and absence of cuticular ornamentation represent the vast majority of the species of the genus. These were grouped based on the relative position of the amphidial fovea, that is, the ratio between the distance of the anterior edge of the amphidial fovea in relation to the anterior end of the body divided by the diameter of the head (Amph ant/hd). For species where the measurements/proportions used in the classification of the groups were not informed in the original description, the measurements/proportions were measured from the drawings available in their respective descriptions. Based on the Amph ant/hd ratio, four other groups were differentiated: Group 4: Amph ant/hd ratio ≤1; Group 5: Amph ant/hd ratio >1–1.5; Group 6: Amph ant/hd ratio >1.5–2 and Group 7: Amph ant/hd ratio >2. Groups 4, 5 and 6 were divided into subgroups. In these subgroups, species whose amphidial fovea occupied 50% or more of the corresponding body region (amphids ≥ 50% cbd) were presented in Subgroup A (4A, 5A, 6A). Those in which the amphidial fovea occupied less than 50% of the corresponding body region (amphids < 50% cbd) formed Subgroup B (4B, 5B, 6B).

In cases where the Amph ant/hd ratio of the species presented variation that could place it in two distinct groups, the group in which the species was inserted was chosen in the following order of priority: if the species had a specific holotype, the measurement

**Table 1 Criteria used in the division of groups and subgroups of valid species of the genus *Microlaimus*.** Distance from the anterior edge of the amphidial fovea to the anterior end of the body divided by the head diameter (Amph ant/hd); Corresponding body diameter (cbd); not applicable (−).

| | Group | Subgroup |
|---|---|---|
| Long spicules (above 2.8 cloacal diameters) | 1 | – |
| Gubernaculum absent | 2 | – |
| Cuticle ornamented with punctuation and/or bars | 3 | – |
| Amph ant/hd ≤1 | 4 | 4A (amphids ≥ 50% cbd) |
| | | 4B (amphids < 50% cbd) |
| Amph ant/hd >1–1.5 | 5 | 5A (amphids ≥ 50% cbd) |
| | | 5B (amphids < 50% cbd) |
| Amph ant/hd >1.5–2 | 6 | 6A (amphids ≥ 50% cbd) |
| | | 6B (amphids < 50% cbd) |
| Amph ant/hd >2 | 7 | – |

obtained from this specimen was used to determine the group; if no holotype was designated, the group was selected according to the measurement interval. For example: if the variation was between 1.2 and 1.6, the species would be included in Group 5 (Amph ant/hd >1 to ≤1.5) because the variation is decreasing when considering the values that delimit the groups, and if it varied from 1.5–1.8, for example, the group would be 6 (Amph ant/hd >1.5 to ≤2), as the variation is increasing in relation to the delimiting value "1.5". When this occurred, observations were made where relevant (see Results and Discussion, 'Specific observations').

## RESULTS AND DISCUSSION

### List of valid *Microlaimus* species

After reviewing the species of *Microlaimus* and the species of other closely related genera, some species transfers between *Bolbolaimus* and *Microlaimus* were proposed.

Transfers were determined based on the presence of enlarged pharyngeal peribuccal tissue in the pharyngeal bulb in *Bolbolaimus vs.* the non-enlarged or slightly enlarged peribuccal tissue in *Microlaimus*. This characteristic was adopted to differentiate *Bolbolaimus* from *Microlaimus* in the identification key provided by *Kovalyev & Tchesunov (2005)* for the Microlaimidae genera. *Tchesunov (2014)* highlighted the same characteristic for distinguishing these genera. We agree with the previously cited manuscripts and consider this to be the most relevant characteristic for differentiating these genera. Other characteristics mentioned in the *Bolbolaimus* diagnosis, such as: head not set off, buccal cavity strongly sclerotized with a large dorsal tooth, copulatory apparatus strongly sclerotized (according to *Tchesunov, 2014*), amphidial fovea situated <1 corresponding body diameter (cbd) from anterior extremity (according to *Leduc, 2016*) are not characteristics exclusive to this taxon, as they are also observed in *Microlaimus* species. Therefore, these characteristics were not used in the present study to establish differences between the cited genera.

The literature on *Bolbolaimus* presents divergent information regarding both the composition and the number of species considered valid for the genus. *Leduc (2016)* and *Long et al. (2017)* considered nine species as valid, but the composition of species in their respective lists presents divergences. The last study in which a species of *Bolbolaimus* was described considers 11 valid species for the genus (*Wen et al., 2023*). According to the *Nemys (2024)* and *WoRMS Editorial Board (2024)* platforms, twelve species are valid. When analyzing this information and evaluating the morphological characteristics of the compiled species, it was possible to observe that in some species the characteristic highlighted as being relevant for diagnosing the *Bolbolaimus* genus (presence of the peribuccal bulb), is absent.

As it does not have a peribuccal bulb, the species *Bolbolaimus brevis Gagarin & Thanh, 2019*, *B. crassiceps* (*Gerlach, 1953*), *B. obesus Long et al., 2017*, *B. parvus Gagarin & Thanh, 2019* and *B. tongaensis Leduc, 2016* were transferred to the genus *Microlaimus*. In addition to these species, we consider *B. abebei Muthumbi & Vincx, 1999* and *B. bahari Muthumbi & Vincx, 1999*, as species of *Microlaimus*. Such transferals were previously proposed by *Tchesunov (2014)*, who stated that these two species are not characterized by a prominent anterior peribuccal bulb of the pharynx and thus, may not differ significantly from species of the genus *Microlaimus* in this respect. Nonetheless, the same species were listed as valid for *Bolbolaimus* (*Leduc, 2016*; *Long et al., 2017*) and as "*uncertain > taxon inquiredum*" in the *Nemys (2024)* and *WoRMS Editorial Board (2024)* platforms. Some of the aforementioned species present a slight dilation of the pharyngeal region that accommodates the oral cavity. However, this slight dilation does not form a perioral bulb. In other species (*B. brevis, B. obesus* e *B. parvus*) the dilation in the anterior region of the pharynx is completely absent.

*Bolbolaimus crassiceps*, originally described in the genus *Microlaimus*, was transferred to *Bolbolaimus* by *Jensen (1978)*. However, we believe that this species should be relocated to the genus *Microlaimus*, since it presents a slight dilation of the anterior portion of the pharynx, which does not form a peribuccal bulb. The presence of a dilation in the anterior region of the pharynx can be observed in other *Microlaimus* species (*e.g., M. acinaces Warwick & Platt, 1973*, *M. campiesis Lima, Neres & Esteves, 2022*, *M.* falciferus *Leduc & Wharton, 2008*, *M. oblongilaimus Gerlach, 1955*, *M. modestus Manoel, Neres & Esteves, 2024b*). However, in these species, such enlargement does not form the muscular structure of a peribuccal bulb (see central image in Fig. 1, *Bolbolaimus* sp.).

For some *Microlaimus* species, the presence of a peribuccal bulb was originally described: *Microlaimus affins Gerlach, 1958* (original description: the pharynx expands anteriorly to form a pharyngeal bulb), *M. conothelis* (*Lorenzen, 1973*), *Jensen, 1978* (original description: bulbous pharyngeal muscles in the buccal cavity region), *M. dimorphus Chitwood, 1937* (original description: stomach region set off from remainder of pharynx and heavily cuticulararized) and *M. robustidens Schuurmans Stekhoven & De Coninck, 1933* (original description: anterior portion of the pharynx embracing the buccal cavity, swollen and strongly muscular). *Chitwood, 1937* also mentions that *Microlaimus dentatus* (transferred to the genus *Pseudomicrolaimus Sergeeva, 1976* by *Kovalyev & Tchesunov, 2005*) and *M. dimorphus* appear to be more closely related to *M. robustindens*

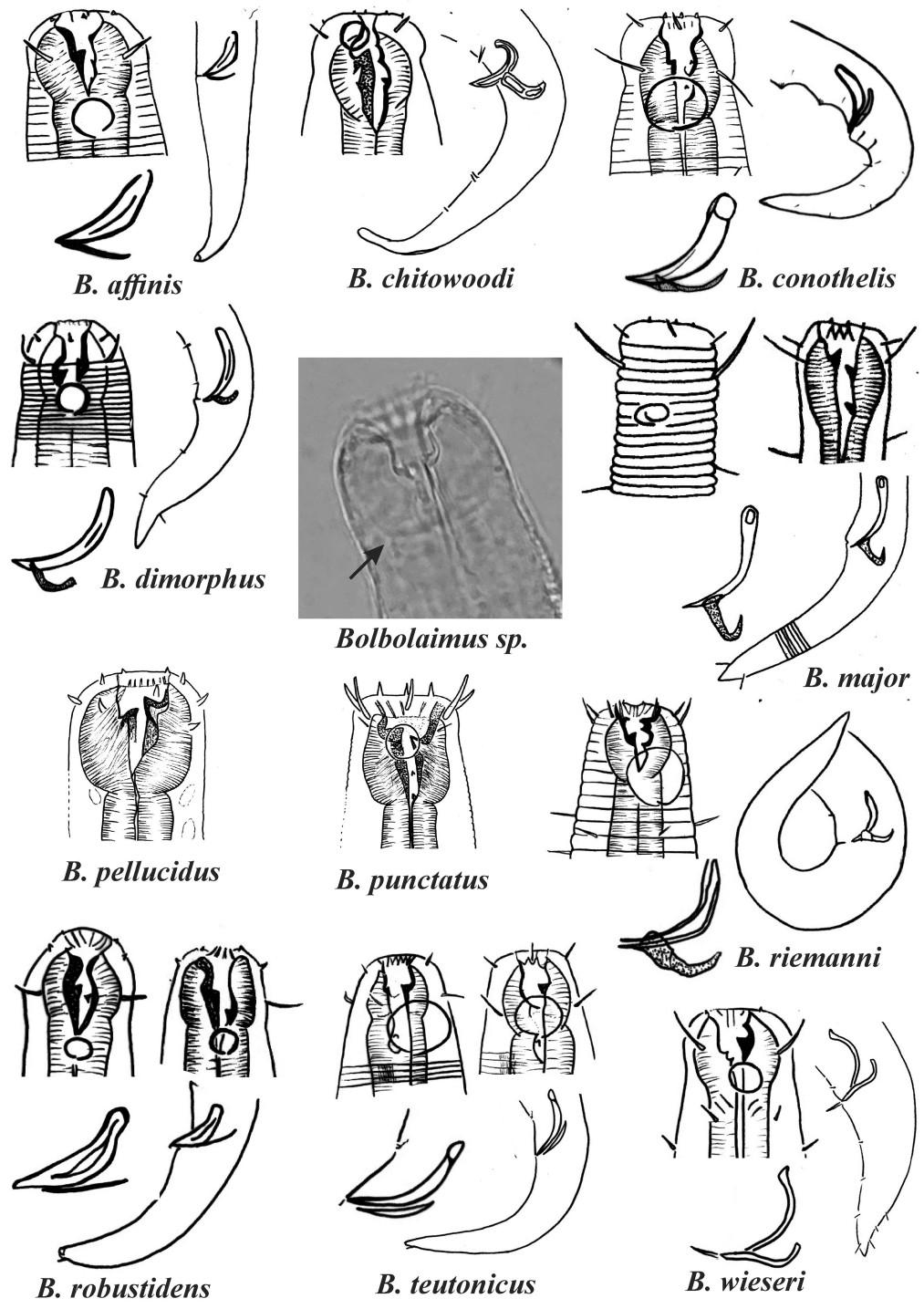

**Figure 1 Drawings of the species considered to belong to the genus *Bolbolaimus* after the review carried out.** The species has the drawing of the anterior region (buccal cavity, cephalic arrangement and amphidial fovea), posterior region (tail, spicule and gubernaculum) and spicule and gubernaculum enlarged. *B. pellucidus* and *B. punctatus* had no posterior body drawings available. Central photo highlights the peribuccal bulb (arrow) characteristic of the genus. All drawings were redone based on original descriptions. Photo credit: Patricia Neres.

than other species of the genus in the following respects: the stomach region is distinctly set off from the remainder of the pharynx and the and the stomach lining and teeth are heavily cuticularized). Therefore, as they present a peribuccal bulb, the species *M. affinis*, *M. conothelis*, *M. dimorphus* e *M. robustindens* were transferred to the genus *Bolbolaimus*.

Wieser (1954) redescribed a species that he initially identified as *M. dimorphus*. However, Hopper (1961) did not considered the specimens described by Wieser as belonging to the species *M. dimorphus*, since the position of the secretory excretory pore differs significantly (the secretory excretory pore is located near the anterior end of the body in *M. dimorphus versus* at the same level as the terminal bulb in the specimens identified by Wieser). As such, the name *M. wieseri* was attributed to the species described in 1954 and, later, Jensen (1978) transferred this species to the genus *Bolbolaimus*. We agree with this transfer, based on the presence of the peribuccal bulb. This fact reinforces our motives for transferring *M. dimorphus* to the genus *Bolbolaimus*, since they are closely related species. Thus, eleven species were considered valid for the genus *Bolbolaimus*, which are represented in Fig. 1.

Finally, *Microlaimus mnazi* (Muthumbi & Vincx, 1999) which was originally described as *Aponema*, were transferred from *Aponema* to *Microlaimus* because of the presence of two testes (Kovalyev & Miljutina, 2009). However, Tchesunov (2014) reestablished the species in the original genus, agreeing with Muthumbi & Vincx, 1999 on the grounds that: (1) gubernaculum as a structural character is more evident and easily observable than male gonads; (2) number of testes is not reported for many, if not the majority, of microlaimid species; (3) posterior testis may be very reduced in some microlaimids (*e.g.*, *Acanthomicrolaimus jenseni* Stewart & Nicholas, 1987) and thus, the presence or absence of the posterior male gonad may have no distinct hiatus for discrimination. Here, we agree with the arguments of Tchesunov (2014) and consider the species in question as *Aponema mnazi*. Thus, the list of *Microlaimus* species resulted in a total of 85 valid species, which are presented in Appendix 1.

## Diagnostic characteristics of *Microlaimus* species

The shape of the setae (papilliform or setiform) of the second and third circles of the cephalic arrangement, the relationship between the length of the cephalic setae and the cephalic diameter, the percentage of the body diameter that the amphidial fovea occupies (Amph%) and its position in relation to the anterior extremity of the body, provided important taxonomic information that was used to distinguish *Microlaimus* species (Lima, Neres & Esteves, 2022; Manoel, Neres & Esteves, 2024b). We consider these characteristics to be the most important for the identification of *Microlaimus* species, and the formation of species groups was based on these characteristics in this study.

Additionally, spicule size (proportion in relation to the cloacal diameter) and morphology, gubernaculum morphology (as well as its presence or absence), the presence or absence of cuticular ornamentation, the composition of the cephalic arrangement (papillae, setiform papillae and setae), the length of the cephalic setae in relation to the cephalic diameter and the morphology of the buccal cavity (number and level of
sclerotization of the teeth), are characteristics that must be considered and carefully evaluated for intraspecific identification.

The presence of hypodermical gland rows, associated or not, with pores or short setae, seen in some *Microlaimus* species, has been highlighted by several authors as a distinctive characteristic for intraspecific identification within this genus (*Hopper & Meyers, 1967*; *Jensen, 1978*; *Muthumbi & Vincx, 1999*; *Manoel, Neres & Esteves, 2024b*). However, it is possible that the occurrence of these structures has not been reported for species with older descriptions. The record of occurrence of hypodermical glands in *Microlaimus* species may be associated with the advancement of optical microscopy.

The presence of supplements (precloacal papillae) can also help in species identification (see Appendix 1). However, it should be noted that these structures can often be difficult to visualize. Precloacal and caudal setae are characteristics that should be considered with caution, as they can be lost during sample processing and specimen preparation. The spicule and gubernaculum morphologies are extremely important for species identification. The creation of an illustrated key facilitates this comparative analysis, which would be difficult to achieve through a descriptive dichotomous key.

*Armenteros, Vincx & Decraemer (2010)* included tail shape and testes number and size as the main interspecific morphological variations found. However, in practice, these characteristics can rarely be used for identification. The tail in most species is conical and in rare situations it can be used as an additional characteristic to differentiate species (*e.g.*, *M. brevis*, *M. nordestinus Manoel, Neres & Esteves, 2024b*, *M. orientalis Gagarin & Thanh, 2011*, *M. parvus*, *M. validus Gagarin & Tu, 2014*). The tail is conical in most Microlaimidae species, and any differences observed are mainly associated with relative tail length (*Kovalyev & Tchesunov, 2005*). Regarding the testes, in addition to being difficult to visualize, in many *Microlaimus* species (approximately 30) information about male gonads is completely absent (*Lima, Neres & Esteves, 2022*).

## Illustrated key of *Microlaimus* species

Group 1 (Fig. 2) includes six species of the genus *Microlaimus* that present long spicules (≥2.8 dc). Among these, the one with the proportionally shortest spicule length is *M. tenuispiculum de Man, 1922* (spicule equivalent to 2.8 times the dc). The species in this group differ from each other in terms of spicule length (see Appendix 1) and gubernaculum morphology (Fig. 2). Additionally, the relative position of the amphids as well as the size of these structures, can help in their identification. It is worth noting that *M. korari Leduc, 2016* presents very faint longitudinal bars, visible at the level of pharyngeal bulb and posteriorly, however, as the spicule of this species is elongated and this characteristic is easier to visualize, it was chosen for the classification of the species in Group 1.

Group 2 (Fig. 2) only comprises two species, *M. minutissimus* (*Kovalyev & Miljutina, 2009*) *Tchesunov, 2014* and *M. nanus Blome, 1982*, both characterized by the absence of the gubernaculum. Differences in the relative position of the amphidial fovea, size and spicule morphology can help in the identification/distinction of these species (Fig. 2).

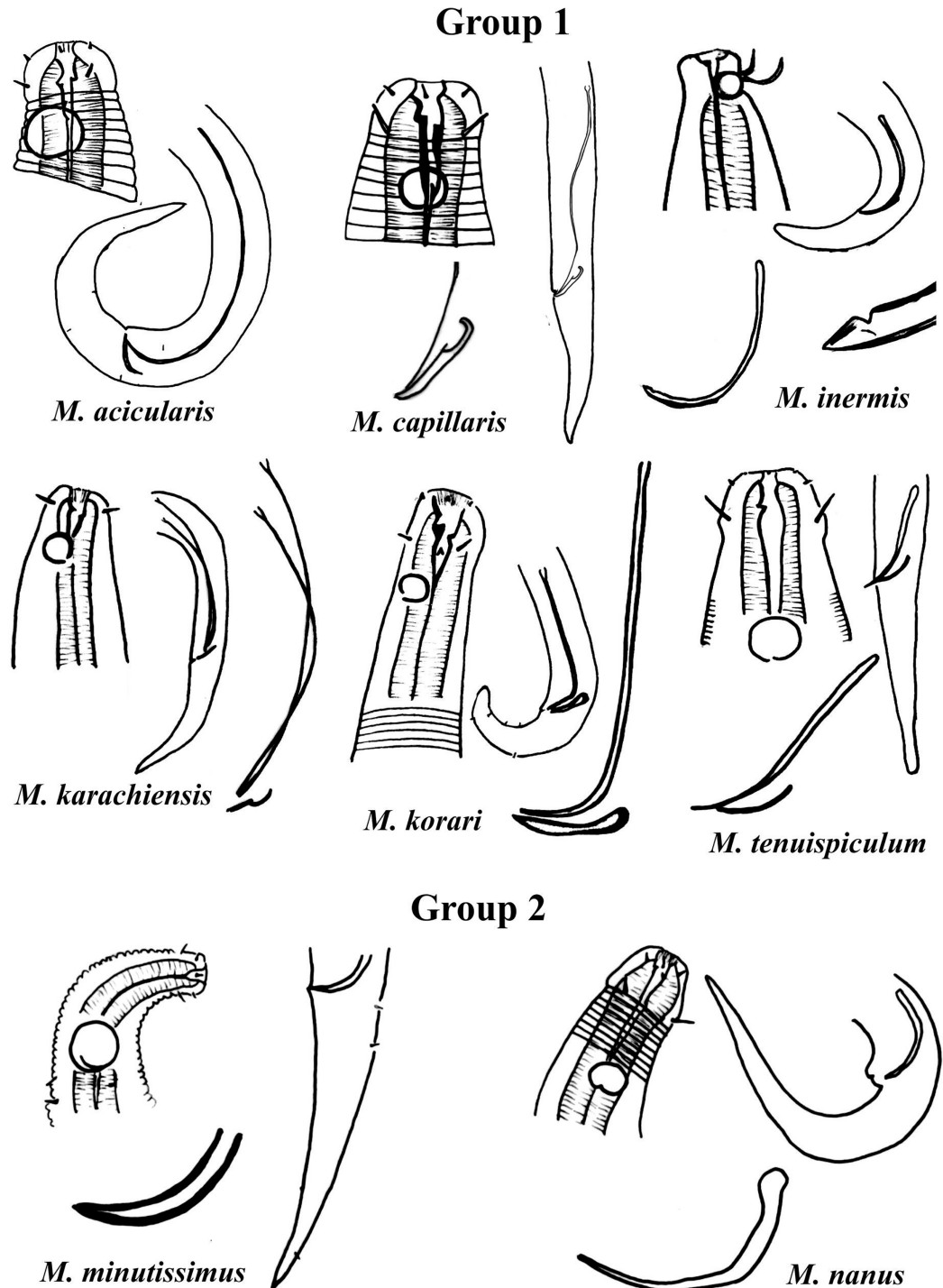

**Figure 2** *Microlaimus* **species of Group 1 and Group 2.** Each species has the drawing of the anterior region (buccal cavity, cephalic arrangement and amphidial fovea), posterior region (tail, spicule and gubernaculum) the spicule and gubernaculum enlarged. *M. capillaris* (only end spicule and gubernaculum enlarged), *M. inermis* (end spicule enlarged below the tail). All drawings were redone based on original descriptions.           

Group 3 (Fig. 3) is formed by *M. annelisae Jensen, 1976*, *M. decoratus de Ward, 1989*, *M. falciferus Leduc & Wharton, 2008*, *M. ostracion Schuurmans Stekhoven, 1935*, *M. punctulatus Gerlach, 1950* and *M. tongaensis Leduc, 2016*, species that present a cuticle with ornate rings (longitudinal bars and/or dots). *M. decoratus*, *M. falciferus* and *M. tongaensis* present longitudinal bars in the cuticle rings. For the latter species, *Leduc (2016)* mentions that the cuticle bars are visible at the level of the nerve ring and in the posterior region of the body. *M. punctulatus* has fine punctuations that appear to be arranged in transverse rows, while *M. annelisae* and *M. ostracion* present punctuations in the annuli of the most anterior part of the cervical region of the body and in the posterior portion of the tail (where the striations are thinner) and longitudinal bars in the annuli of the other parts of the body. These species are also differentiated by the position of the amphidial fovea in relation to the anterior end of the body. *M. annelisae* and *M. punctulatus* present amphids further away from the anterior end of the body (Amph ant/hd close to 1.5). This same proportion in *M. falciferus* and *M. tongaensis* is less than 1.0 and in *M. ostracion* it is around 1.0. *M. annelisae* is easily differentiated by having large amphids, which occupy around 80% of the corresponding body diameter. In *M. punctulatus* this structure occupies between 40–65%, in *M. falciferus* between 35–45% and in *M. ostracion* around 1/3 of the corresponding body diameter. Differences in spicule and gubernaculum morphologies can also be considered when identifying the species (Fig. 3).

Group 4 (Figs. 4–6) comprises the largest number of species, 27 in total. These species are characterized through the positioning of the amphidial fovea relatively closer to the anterior end of the body (Amph ant/hd ≤ 1). Subgroup 4A (A-amphidial fovea greater than or equal to 50% of the corresponding body diameter) is formed by 12 species and Subgroup 4B (B-amphidial fovea less than 50% of the corresponding body diameter) by 15 species. In each of these subgroups, the species can be differentiated by the composition of the cephalic arrangement (papillae, septiform papillae, setae) and by the relative length of the cephalic setae (using the cephalic diameter as a proportion factor) (Appendix 1). Additionally, other characteristics, such as the morphology of the spicules, of the gubernaculum and of the buccal cavity (observing the size and degree of sclerotization of the teeth) are important for distinguishing these species.

In Subgroup 4A (Figs. 4 and 5), the longest spicules are found in *M. abebei* (2–2.3 dc), *M. bahari* (1.9–2 dc) and *M. lunatus* (*Wieser & Hopper, 1967*) *Jensen, 1978* (1.7 dc). The gubernaculum is distinctly different in *M. nympha* (*Bussau & Vopel, 1999*) *Tchesunov, 2014*, whose mouth cavity is unarmed. The teeth are more prominent (developed/sclerotized) in *M. acinaces*, *M. amphidius Kamran, Nasira & Shahina, 2009*, *M. papillatus Allgén, 1959* and *M. paraconothelis Kovalyev & Tchesunov, 2005*. The presence of supplements (precloacal papillae) may also help in the differentiation/identification of species in Subgroup 4A (see Appendix 1). Among the species presented in Subgroup 4B (Figs. 5 and 6), the teeth appear to be comparatively smaller in *M. discolensis Bussau, 1993* and *M. falklandiae Allgén, 1959*; the spicules are larger in *M. obesus* and *M. orientalis* and described as "weakly cuticularized" in *M. arenicola Schulz, 1938*; the gubernaculum stands out for presenting a distinct morphology in *M. crassiceps*, *M. vitorius Lima, Neres & Esteves, 2022* and *M. undulatus*. The presence of hypodermic glands associated with pores

## Group 3

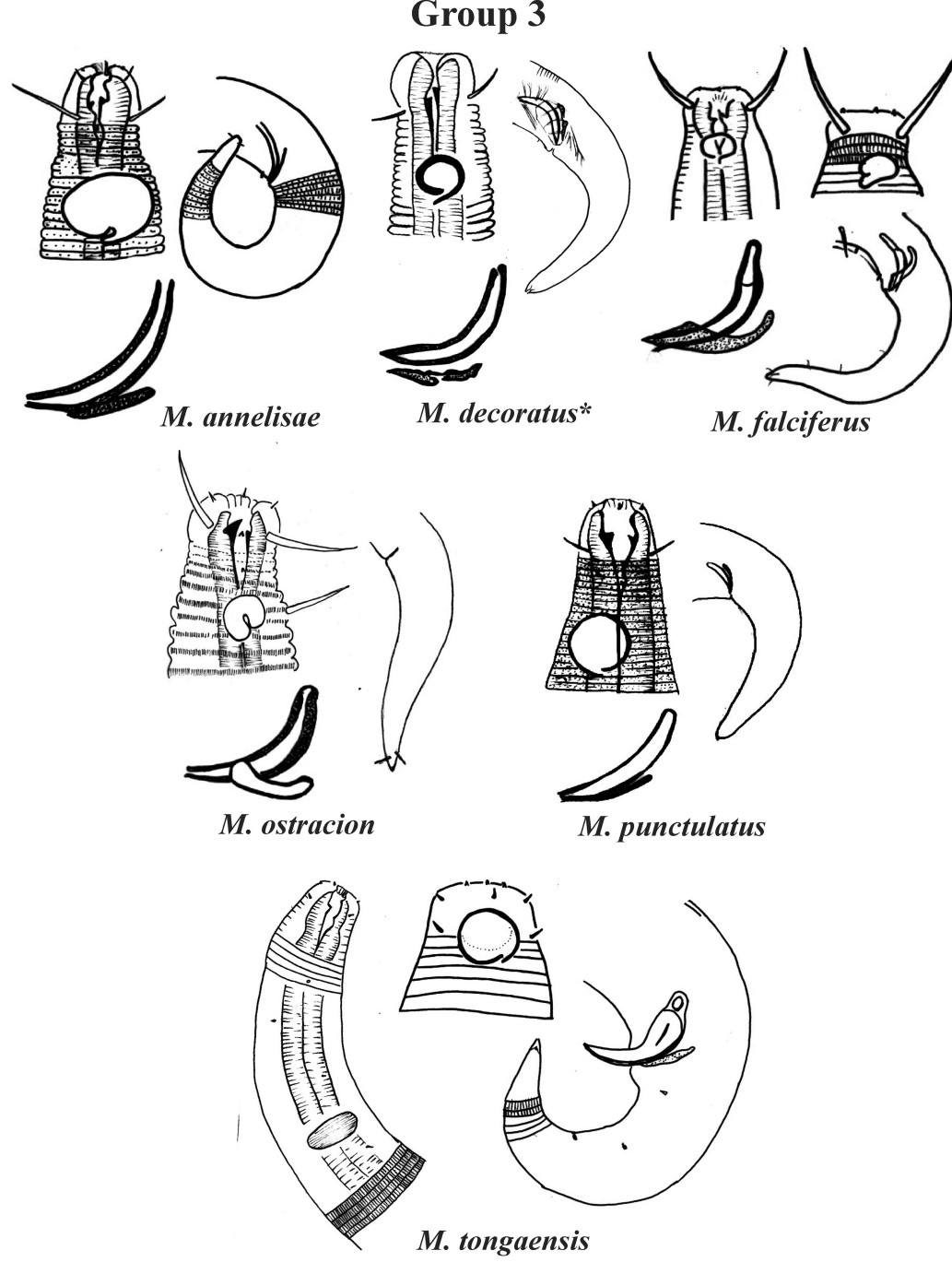

**Figure 3 *Microlaimus* species of Group 3.** Each species has the drawing of the anterior region (buccal cavity, cephalic arrangement, amphidial fovea and cuticle ornamentation), posterior region (tail, spicule and gubernaculum), and spicule and gubernaculum enlarged. *M. decoratus* (*the cuticle ornamentation is not present in the drawing, but is mentioned in the text). *M. ostracion* (drawing based on *Jensen, 1976*; drawing of tail female). All drawings were redone based on original descriptions.

## Group 4: Subgroup A

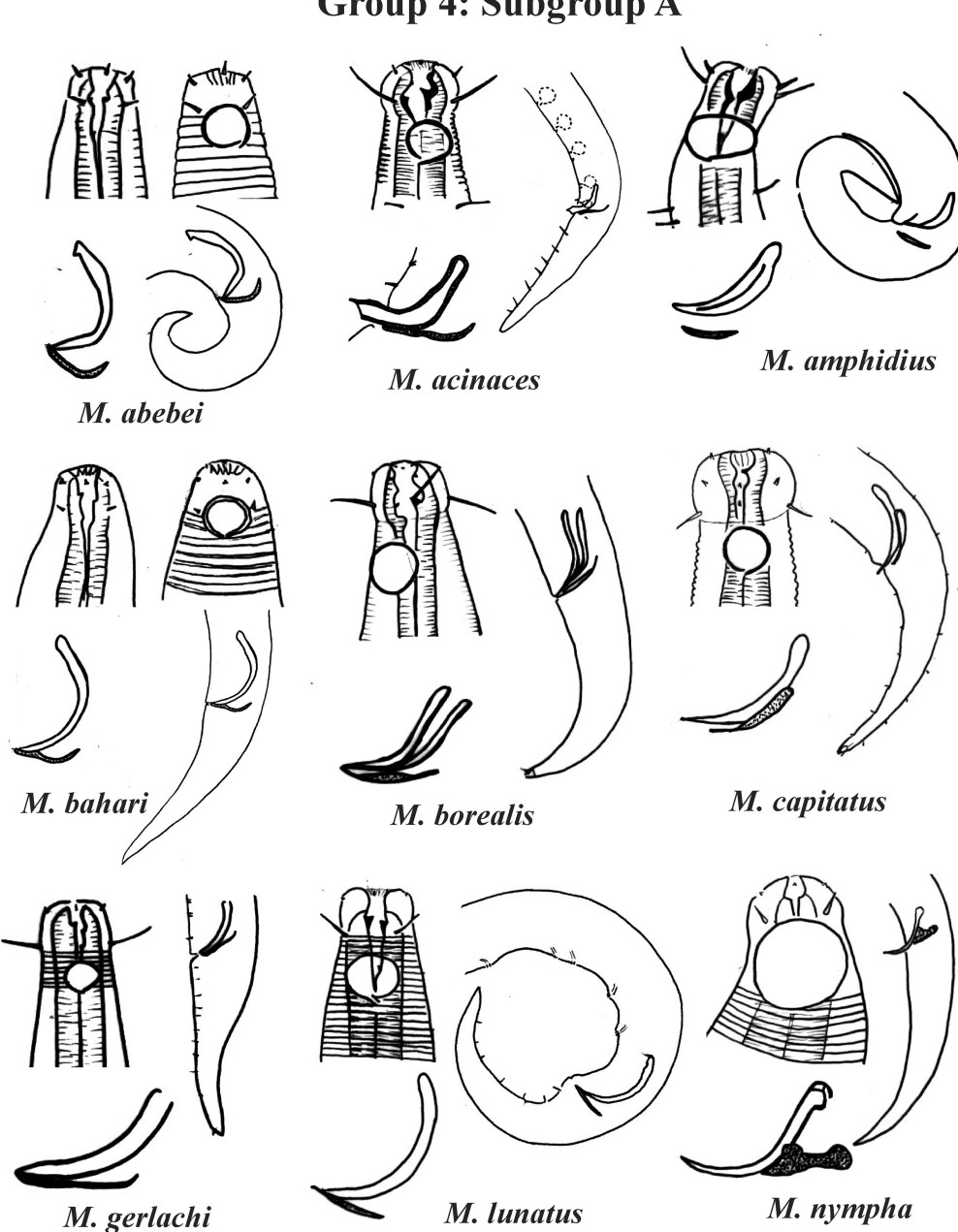

**Figure 4** ***Microlaimus*** **species of Group 4 (Subgroup A) (in part).** Each species has the drawing of the anterior region (buccal cavity, cephalic arrangement and amphidial fovea), posterior region (tail, spicule and gubernaculum) and spicule and gubernaculum enlarged. M. *acinaces, M. gerlachi* and *M. lunatus* have the precloacal supplements illustrated. All drawings were redone based on original descriptions.

was described for *M. discolensis* (Appendix 1). For both subgroups (4A and 4B), the presence or absence of precloacal and/or caudal setae (Appendix 1) are distinctive features that should be considered with caution, since these structures may be lost during sample processing and specimen preparation.

## Group 4: Subgroup A (cont.)

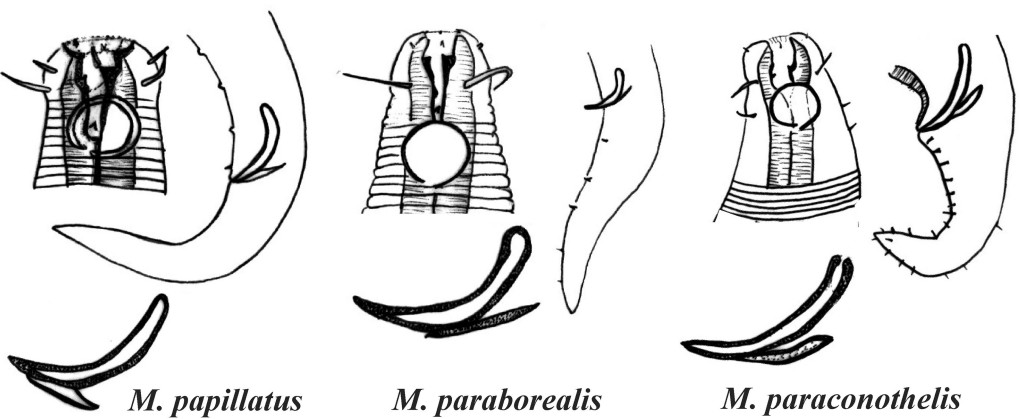

M. papillatus      M. paraborealis      M. paraconothelis

## Group 4: Subgroup B

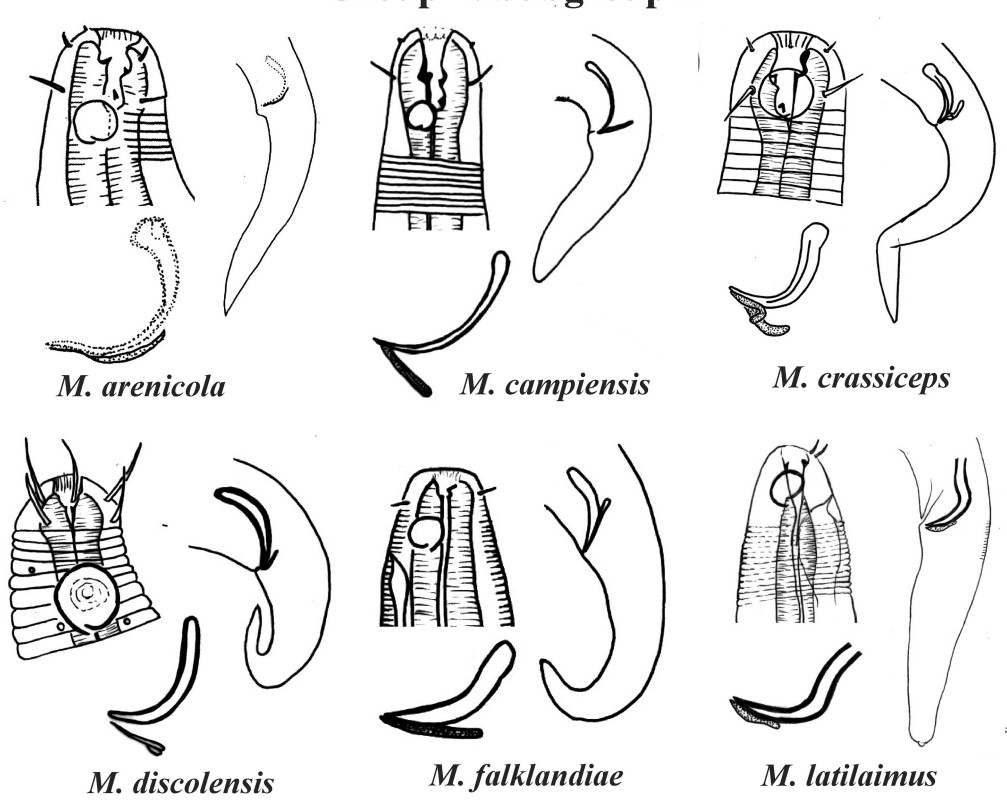

M. arenicola      M. campiensis      M. crassiceps

M. discolensis      M. falklandiae      M. latilaimus

**Figure 5** *Microlaimus* **species of Group 4 (Subgroup A) (continuation) and Group 4 (Subgroup B) (in part).** Each species has the drawing of the anterior region (buccal cavity, cephalic arrangement and amphidial fovea), posterior region (tail, spicule and gubernaculum) and spicule and gubernaculum enlarged. *M. papillatus* has the precloacal supplements illustrated. M. paraborealis (drawing based on *Gerlach, 1950*). *M. arenicola* (drawing based on *Blome, 1982*). All drawings were redone based on original descriptions.

## Group 4: Subgroup B (cont.)

*M. naidinae*          *M. obesus*          *M. orientalis*

*M. paraaffinis*          *M. parvus*          *M. pinguis*

*M. sensus*          *M. vitorius*          *M. undultus*

**Figure 6** ***Microlaimus* species of Group 4 (Subgroup B) (continuation).** Each species has the drawing of the anterior region (buccal cavity, cephalic arrangement and amphidial fovea), posterior region (tail, spicule and gubernaculum) and spicule and gubernaculum enlarged. All drawings were redone based on original descriptions.

Group 5 (Figs. 7–9) consists of 19 species, which have the ratio Amph ant/hd >1 to 1.5 in common. As in the previous group, these species were subdivided into two subgroups (5A and 5B), adopting the same criterion used to divide Subgroups 4A and 4B (percentage of the corresponding body diameter that the amphidial fovea occupies). Subgroup 5A

## Group 5: Subgroup A

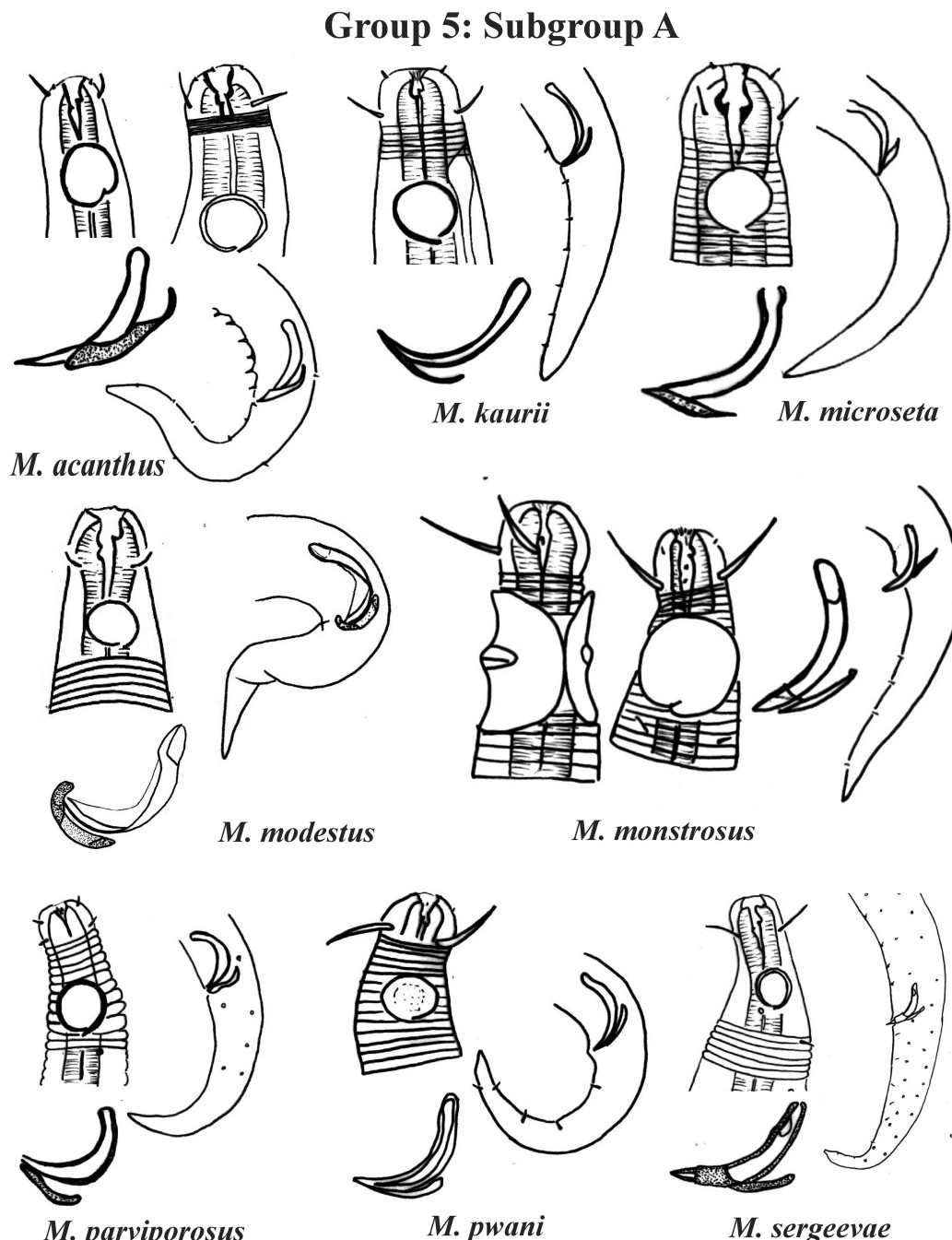

**Figure 7 *Microlaimus* species of Group 5 (Subgroup A).** Each species has the drawing of the anterior region (buccal cavity, cephalic arrangement and amphidial fovea), posterior region (tail, spicule and gubernaculum) and spicule and gubernaculum enlarged. *M. acanthus* has the precloacal supplements illustrated (variation found in fovea position 1.1–1.8 hd, fits into Subgroups 5A and 6A, see details in 'Specific observations'). *M. parviporosus* has cuticular pores illustrated in tail. *M. sergeevae* has precloacal supplements and cuticular pores illustrated in posterior region. All drawings were redone based on original descriptions.

## Group 5: Subgroup B

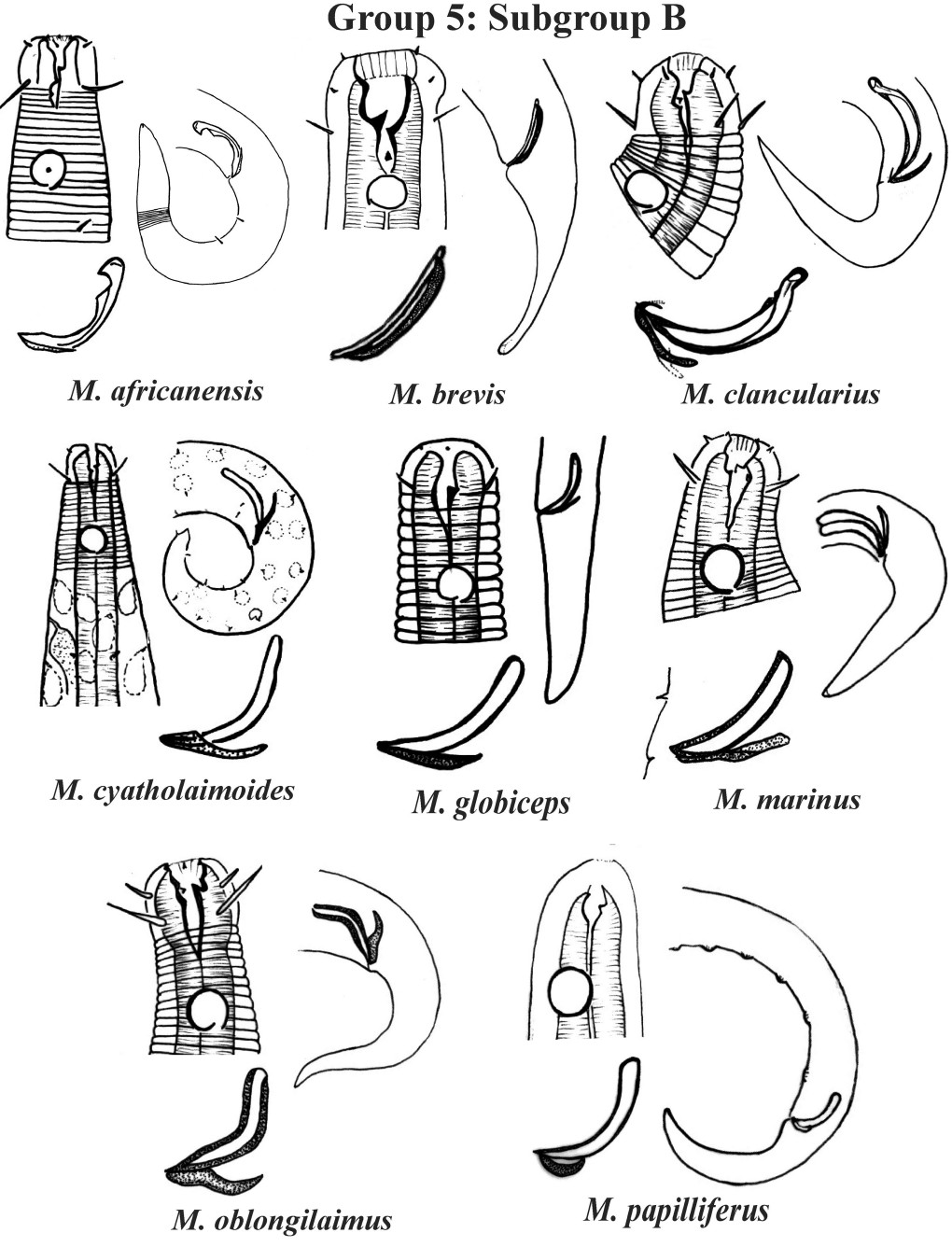

**Figure 8** ***Microlaimus* species of Group 5 (Subgroup B) (in part).** Each species has the drawing of the anterior region (buccal cavity, cephalic arrangement and amphidial fovea), posterior region (tail, spicule and gubernaculum) and spicule and gubernaculum enlarged. *M. cyatholaimoides* has cuticular pores illustrated. *M. papilliferus* has precloacal supplements Illustrated. *M. globceps* (drawing based on *Gerlach, 1950*). *M. marinus* (tail based in *Schuurmans Stekhoven & De Coninck, 1933*). All drawings were redone based on original descriptions.

(Fig. 7) (amphids ≥ 50% of the corresponding body diameter) comprises eight species and Subgroup 5B (Figs. 8 and 9) (amphids < 50% of the corresponding body diameter) comprises 11 species. To differentiate/identify the species of these subgroups, the same

## Group 5: Subgroup B (cont.)

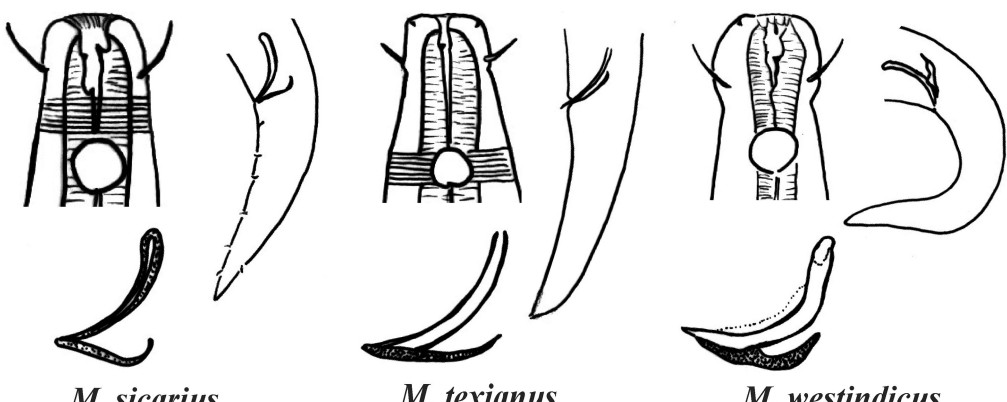

| | | |
|:---:|:---:|:---:|
| *M. sicarius* | *M. texianus* | *M. westindicus* |

**Figure 9** ***Microlaimus* species of Group 5 (Subgroup B) (continuation).** Each species has the drawing of the anterior region (buccal cavity, cephalic arrangement and amphidial fovea), posterior region (tail, spicule and gubernaculum) and spicule and gubernaculum enlarged. *M. texianus* (drawing based on *Wieser, 1954*). All drawings were redone based on original descriptions.

characters mentioned above can also be applied (composition of the cephalic arrangement; relative length of the cephalic setae; spicules and gubernaculum morphology). Allocated to Subgroup 5A, the species *M. acanthus* (*Jayasree & Warwick, 1977*) *Kovalyev & Tchesunov, 2005*, whose holotype was not designated, presented a variation in the Amph ant/hd ratio (between 1.1 and 1.8) (manuscript presents a plate with the drawing of the heads and tails of two males). Although *M. acanthus* presents this variation in the relative position of the amphidial fovea, the species presents characteristic precloacal supplements that can be used for its identification, since, apparently, such structures are easy to visualize (4 to 6 precloacal papillae, with a robust seta 3–4 μm on each papilla) (Fig. 7; Appendix 1). In this same subgroup, *M. monstrosus Gerlach, 1953* can be easily identified due to the size of the amphids, which occupy the entire diameter of the corresponding region of the body. The presence of cuticular pores in four submedian rows was mentioned for *M. parviporosus Miljutin & Miljutina, 2009* and *M. sergeevae Revkova, 2020*. In Subgroup 5B, *M. africanensis Furstenberg & Vincx, 1992* and *M. clancularius Bussau, 1993* have spicules with shapes that are easily distinguishable from the other species in the group. *M. brevis*, in addition to having a gubernaculum of similar length to the spicule, an unusual characteristic for the genus, has a tail with a distinct morphology (conical-cylindrical that gradually narrows) when compared to the other species in the genus. Hypodermal glands arranged in four rows associated with small setae were described for *M. cyatholaimoides de Man, 1922* (Fig. 8; Appendix 1).

Group 6 (Figs. 10–12) consists of 17 species, which present an Amph ant/hd ratio >1.5 to 2. Applying the same criterion used previously (percentage of the corresponding diameter of the body occupied by the amphid), Subgroups 6A (seven species) and 6B (11 species) were formed. In both subgroups, most species present a buccal cavity with

## Group 6: Subgroup A

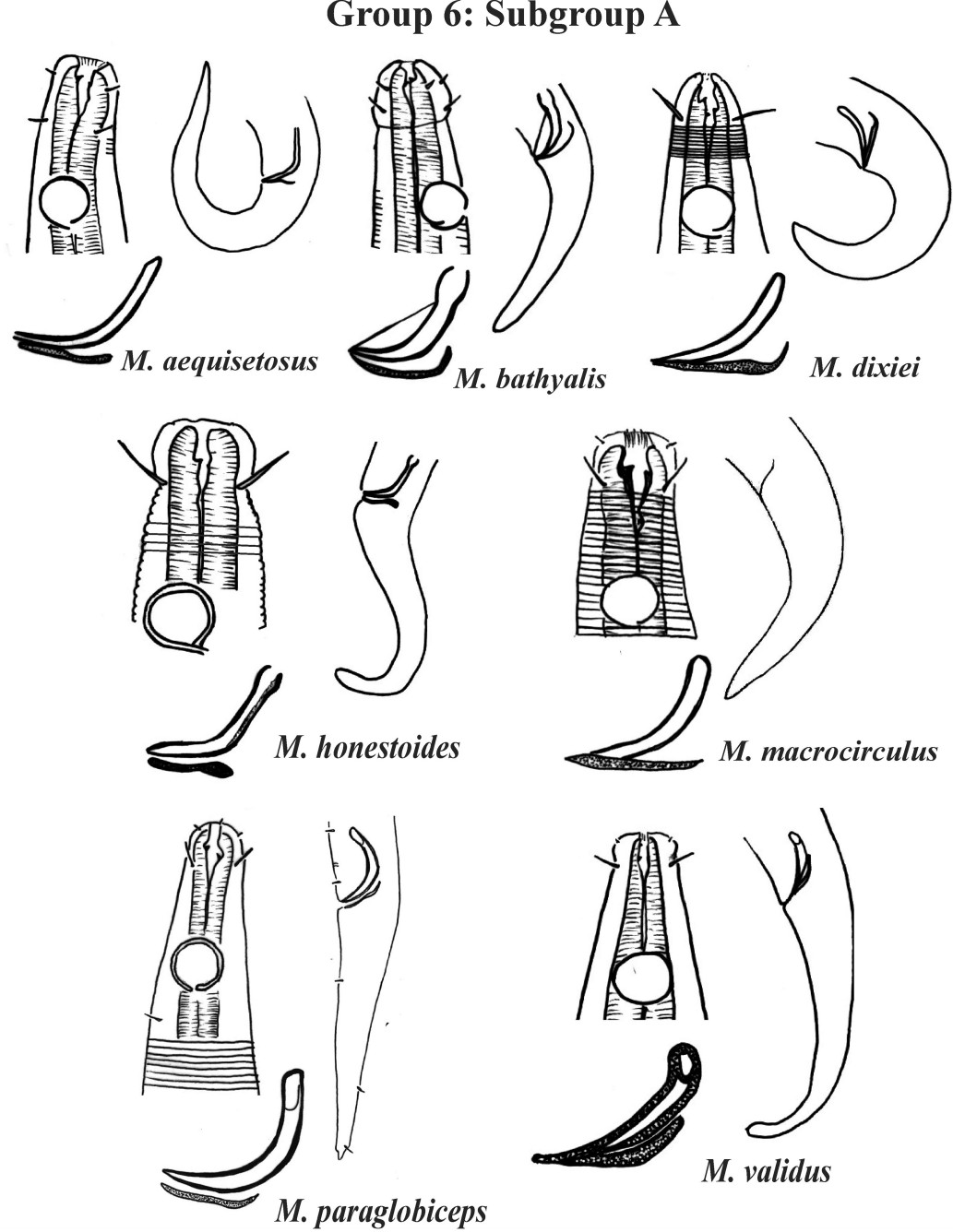

**Figure 10** *Microlaimus* **species of Group 6 (Subgroup A).** Each species has the drawing of the anterior region (buccal cavity, cephalic arrangement and amphidial fovea), posterior region (tail, spicule and gubernaculum) the spicule and gubernaculum enlarged. All drawings were redone based on original descriptions.                               

small teeth, with the exception of *M. macrocirculus Gerlach, 1950* (6A) and *M. formosus Gerlach, 1957* (6B). In Subgroup 6A (Fig. 10), *M. bathyalis* (*Kovalyev & Miljutina, 2009*) *Tchesunov, 2014* can be identified by its strongly highlighted cephalic region (set off), circle

## Group 6: Subgroup B

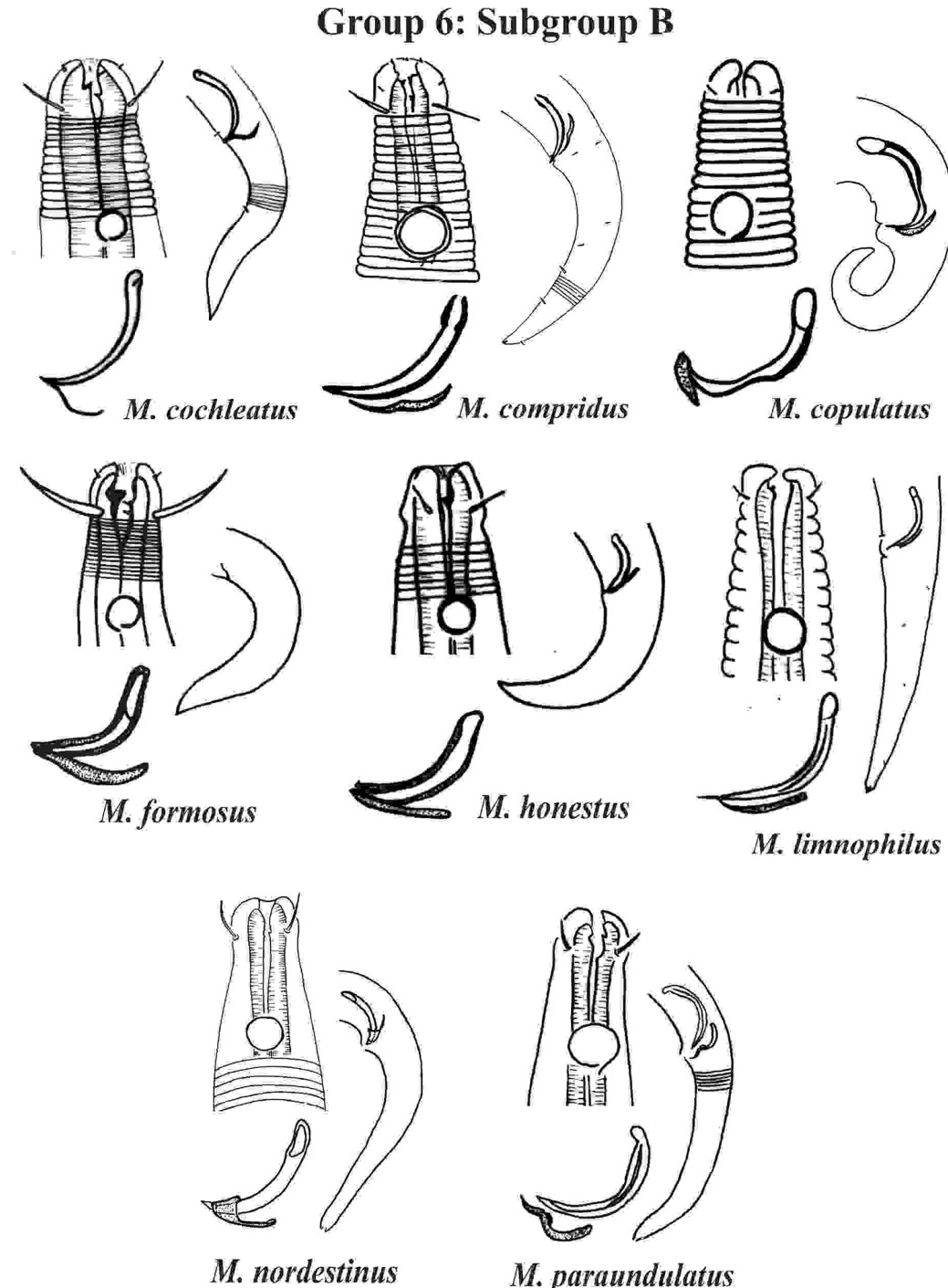

**Figure 11 _Microlaimus_ species of Group 6 (Subgroup B) (in part).** Each species has the drawing of the anterior region (buccal cavity, cephalic arrangement and amphidial fovea), posterior region (tail, spicule and gubernaculum) and spicule and gubernaculum enlarged. _M. compridus_ (drawing based on _Gourbault & Vincx, 1988_). _M. formosus_ (original description, figure caption does not indicate whether the head and tail were of the male or the female). _M. honestus_ (drawing based on _Schuurmans Stekhoven & De Coninck, 1933_). All drawings were redone based on original descriptions.

## Group 6: Subgroup B (cont.)

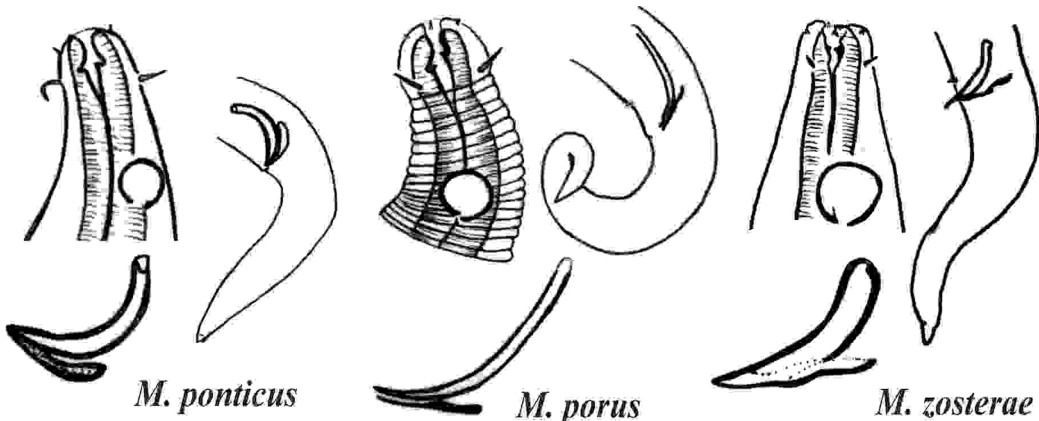

*M. ponticus*   *M. porus*   *M. zosterae*

**Figure 12 *Microlaimus* species of Group 6 (Subgroup B) (continuation).** Each species has the drawing of the anterior region (buccal cavity, cephalic arrangement and amphidial fovea), posterior region (tail, spicule and gubernaculum) and spicule and gubernaculum enlarged. *M. porus* (drawing of the female's head). *M. zosterae* (drawing based on *Kovalyev & Tchesunov, 2005*). All drawings were redone based on original descriptions.               

of short cephalic setae (1.5–2 µm) and by the morphology of the male reproductive system (spicules and gubernaculum). *M. aequisetosus Blome, 1982* and *M. honestoides Meyl, 1954* have curved L-shaped spicules, and can be differentiated from each other mainly by cephalic setae length (Fig. 10). In Subgroup 6B (Figs. 11 and 12), *M. copulatus Jensen, 1988* has spicules with an unusual and peculiar morphology (spicule with a cylindrical proximal portion that is narrow and curved in the middle and thin in the distal portion). The gubernaculum of *M. nordestinus* and *M. paraundulatus Manoel, Neres & Esteves, 2024b* are easily distinguishable from the other species in Subgroup 6B. *M. formosus*, and in addition to having well-developed teeth, have comparatively long cephalic setae (>1 hd in length).

Group 7 (Fig. 13) includes seven species, which present the Amph ant/hd ratio > 2. Within this group, these species can be distinguished by characteristics such as the composition of the cephalic arrangement, the cephalic setae length and the male reproductive apparatus (spicules and gubernaculum). *M. alexandri Lima, Neres & Esteves, 2022* has a buccal cavity with five teeth, two dorsal and three ventrosublateral, and the amphidial fovea occupies the entire corresponding diameter in males (sexual dimorphism of the amphid, females have a fovea <50% of the cbd) (Appendix 1). *M. martinezi (Miljutin & Miljutina, 2009) Tchesunov, 2014* and *M. minutus Muthumbi & Vincx, 1999* present the 3rd circle formed by setiform papillae (Fig. 12). *M. tenuicollis (Gerlach, 1952) Jensen, 1978* can be differentiated by the position of the fovea, and represents the species that comparatively presents amphids furthest from the anterior end of the body (Amph ant/hd = 4.3–4.5) (Fig. 13).

## Group 7

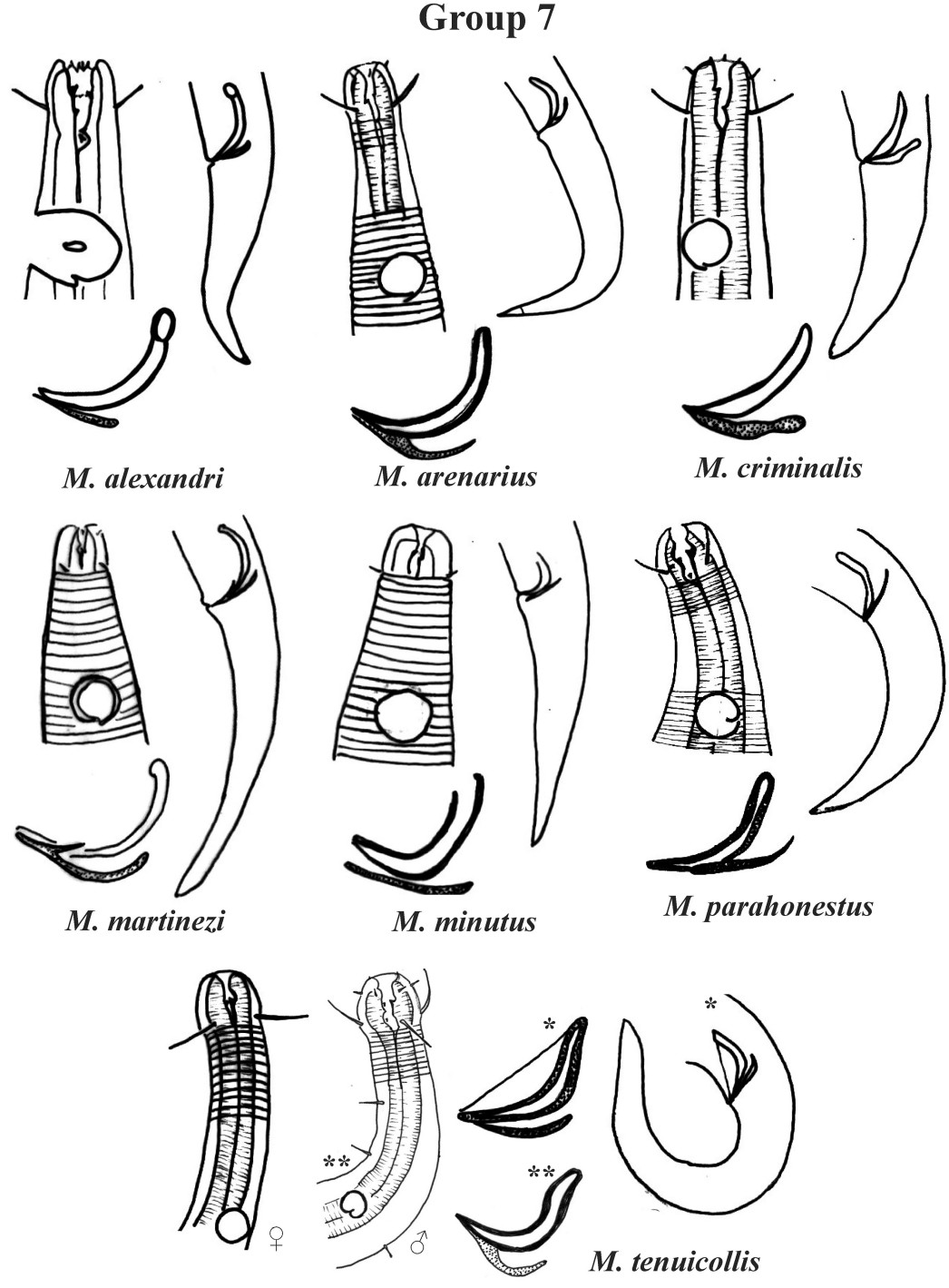

**Figure 13 *Microlaimus* species of Group 7.** Each species has the drawing of the anterior region (buccal cavity, cephalic arrangement and amphidial fovea), posterior region (tail, spicule and gubernaculum) and spicule and gubernaculum enlarged. *M. tenuicollis* (*Gerlach, 1952* original description-only female; *Gerlach, 1953-spicule and tail; **Blome, 198–male head and spicule). All drawings were redone based on original descriptions.

## Specific observations

Regarding the groups formed, some observations should be highlighted: in the *M. macrocirculus* description, it was mentioned that the amphids are located at a distance of 1.5 hd from the anterior end, but in the figure (in original article) it is possible to clearly observe a divergence, since when measuring the proportion using the drawing, the value obtained is $1.8 \times$ hd. In this situation, the value of the proportion measured from the figure (in original article) was considered when choosing the group in which *M. macrocirculus* was inserted. In *M. acanthus*, no holotype was designated. The original description and the drawing provided include an illustration of two male specimens. In this species, the variation found in the position of the amphids among the described specimens is 1.1–1.8 hd, which is an example of a species that could be allocated to two groups (5 or 6; Subgroup 5A or 6A). This species was inserted into Group 5, Subgroup 5A, because the individual for which the head and tail were represented in the original description drawings (male 2 in *Jayasree & Warwick, 1977*) was chosen here for the species classification in its respective group, but the drawing of the head of male 1 was also represented on the plate of subgroup 5A. A similar situation was observed in *M. paraglobceps Revkova, 2017* and *M. nordestinus*, where the data referring to the respective holotypes were used to classify the species into groups, but paratype measurements are divergent (*M. paraglobceps* $1.9 \times$ hd in the holotype and $2.3 \times$ hd in the paratype; *M. nordestinus* $1.9 \times$ hd in the holotype and the variation of $1.6–2.1 \times$ hd between the paratypes).

If a researcher trying to identify a *Microlaimus* species, finds themselves in a similar situation, they should consider more than one of the groups in their analysis and the previously mentioned characteristics should be evaluated, to identify the specimens in question. The same applies to the proportion of the corresponding diameter that the amphid occupies, resulting in the analysis of Subgroups A and B when the variation does not fall completely into a single group.

## Future implications

Discovering and describing the species that inhabit the Earth continues to be a fundamental mission of the discipline of Biology, especially in the face of so many concerning environmental issues (invasive species, climate change, habitat destruction, loss of biodiversity), the need for taxonomic information is greater than ever (*Zhang, 2011*).

Nematoda taxonomy often has a controversial history, not only as a result of the development of procedures in systematics, but also because many nematologists have not produced detailed classifications (*De Ley, Decraemer & Eyualem-Abebe, 2006*). Furthermore, Nematoda is considered a difficult group to identify, mainly due to its small body size. Therefore, the identification of its biodiversity requires the use of optical microscopes, as well as extensive knowledge of forms described in specialized taxonomic literature (*Blaxter & Floyd, 2003*; *De Ley et al., 2005*).

Due to these identification difficulties, most ecological work involving the Phylum is restricted to taxonomic identification at the genus level, failing to reveal the real diversity of the locations studied, in addition to making it impossible to study the worldwide

distribution of species, habitat preferences, as well as knowledge of which species are more resistant or vulnerable to environmental changes.

It is already well-known that cosmopolitanism in most genera of Nematoda does not necessarily apply to all species of the genus (*Zeppilli, Vanreusel & Danovaro, 2011*); and studies suggest the presence of large differences in species distribution at local and regional scales (*Danovaro et al., 2009*; *Lima, 2016*; *Miljutin et al., 2010*; *Vermeeren, Vanreusel & Vanhove, 2004*). *Microlaimus* is a cosmopolitan genus, we cannot affirm this for its species, since most studies do not identify the genus at the species level (*Lima, 2016*).

Therefore, developing a tool that reduces the difficulty of identifying the species of a genus that is commonly found in different types of habitats and depths will help to narrow this gap. And who knows, if several studies with similar aims are published, the distribution patterns, habitat preferences and tolerance to environmental changes, among others, of marine nematodes, one of the most diverse and abundant groups on the planet, will be described at the species level.

## CONCLUSIONS

*A priori*, species can be divided into smaller groups according to metric characteristics or by the absence or presence of a given character. This criterion is objective, which makes comparison relatively more practical. On the other hand, within groups of species, morphological variation (spicules and gubernaculum) gains importance in distinguishing similar species. However, the morphological description of a structure is more subjective, therefore, the same structure can be described differently depending on the taxonomist's point of view, which makes it more difficult to standardize the nomenclature used. Consequently, the construction of a common identification key (using only character descriptions) would be difficult to develop and, above all, difficult to apply in specific identification. Therefore, comparisons using images are more effective in highlighting the differences in these structures between morphologically close species.

*Microlaimus* is known to be a very diverse genus, characterized by species that are generally minute in size and that differ from each other in very subtle characteristics. Therefore, by providing a more practical tool for species identification, this research can be used in several studies (ecological, experimental tests, biological characterization of areas, or even genetic, such as in the study of cryptic species, among others), allowing future studies related to the distribution of species and their interactions with the environment to be performed.

### Funding

Alex Manoel was supported by a FACEPE graduate scholarship: IBPG-1516-2.00/21. The funders had no role in study design, data collection and analysis, decision to publish, or preparation of the manuscript.

## Grant Disclosures

The following grant information was disclosed by the authors:
FACEPE graduate scholarship: IBPG-1516-2.00/21.

## Competing Interests

The authors declare that they have no competing interests.

## Author Contributions

- Andre M. Esteves conceived and designed the experiments, performed the experiments, analyzed the data, authored or reviewed drafts of the article, and approved the final draft.
- Alex Manoel conceived and designed the experiments, performed the experiments, analyzed the data, prepared figures and/or tables, authored or reviewed drafts of the article, and approved the final draft.
- Patricia F. Neres conceived and designed the experiments, performed the experiments, analyzed the data, prepared figures and/or tables, authored or reviewed drafts of the article, and approved the final draft.

## Data Availability

All data are available in the Supplemental File.

## Supplemental Information

Supplemental information for this article can be found online at http://dx.doi.org/10.7717/peerj.19611#supplemental-information.

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
