# Peer review of "Review of the genus Microlaimus de Man, 1880 with an illustrated guide for species identification"

_PeerJ, doi:10.7717/peerj.19611_

## Round 0.1 · original submission · Minor Revisions

Dear Dr. Esteves, I kindly ask you to make the necessary changes so that the reviewers can approve the publication of this article.

Reviewer 1 ·

Basic reporting

The topic of this scientific work is very interesting and relevant for specialists in the field of zoology and parasitology. The review of the genus Microlaimus de Man, 1880, which was made by the authors, includes very important information regarding the morphological characteristics of the genus and the species that belong to it. A thorough analysis of the literature regarding the lists of species of this genus, which have differences, was carried out. Groups of species of the genus Microlaimus with certain characteristics were created, which were presented in an an illustrated guide for species identification of representatives of the genus Microlaimus. However, there are some comments.
In «INTRODUCTION» the authors provide information from various sources on the structure of the family Microlaimidae (number of genera and species), as well as the number of species of the genus Microlaimus. The authors point out the gaps in this area of research and the need to revise the genera of this family. It is also necessary to add brief information on the habitat, distribution of these nematodes and their role in the ecosystem to this section.

Experimental design

The Survey Methodology is consistent with a comprehensive, unbiased coverage of the subject. Sources are adequately cited. The review is organized logically into coherent sections and subsections. However, there are some comments.
The list of references contains more than 50% of outdated literary sources (from 1920 to 1999). It is necessary to reduce the number of outdated sources and add new sources.
Subsection «IV. Diagnostic characteristics of Microlaimus species» - includes basic information regarding the species characteristics of the genus, on the basis of which the illustrated guide was created. Therefore, I advise moving it to the beginning of the work, before the second subsection «II. Illustrated key of Microlaimus species» of the section «RESULTS AND DISCUSSION».
The figures were made according to the numbering of the groups. But figure 4 contains only partially the species of Microlaimus of group 4, subgroup A. And the rest of this subgroup A is placed in figure 5. Also in figure 5 are placed some of the species of Microlaimus of group 4, which belong to subgroup B, and the rest of the species of this subgroup B are in figure 6. The same applies to the distribution of species of groups 5 and 6. I ask the authors to explain this distribution in the figures.

Validity of the findings

There is a well developed and supported argument that meets the goals set out in the Introduction. But there is a remark to the conclusion. In «CONCLUSIONS» section, it is necessary to provide more details on the authors' contributions to the research regarding the review of the genus Microlaimus de Man, 1880. It is also necessary to clarify which areas of research on this topic are promising for future study.

Additional comments

The figures show only parts of the nematode's body. Therefore, this clarification must be made in the figure's title, for example, a cephalic region.
In the titles of the figures, it is necessary to add all the characteristics (for example, spicule, gubernaculum, amphidial fovea, cuticle) that the authors describe in the text and to designate them with abbreviations in the figures of the species.
Line 190. The authors point to a certain characteristic, according to which some species of the genus Microlaimus (M. affinis, M. conothelis and M. dimorphus) were transferred to the genus Bolbolaimus. But above on lines 183-185 the authors give an original description of this characteristic (the presence of a peribuccal bulb) in four species, namely: M. affinis, M. conothelis, M. dimorphus and M. robustindens. Therefore on line 190 it is necessary to add the species M. robustindens.
Line 228-230. The authors indicate that group 3 includes the following species: M. annelisae Jensen, 1976, M. decoratus Pastor de Ward, 1989, M. falciferus Leduc & Wharton, 2008, M. ostracion Schuurmans Stekhoven, 1935 and M. punctulatus Gerlach, 1950. However, below on line 231, another species, M. tongaensis, is also mentioned. Therefore, it is necessary to add the species M. tongaensis to the general list of species in this group on lines 228-230.
I ask authors to pay attention to punctuation throughout the text. There should be a comma after the author's last name and before the year in parentheses. For example, in line 151: (Wen et al., 2023).

Reviewer 2 ·

Basic reporting

The manuscript is clearly written, employing a professional level of English that conveys the content effectively. However, some sentences would benefit from slight rephrasing to enhance readability and eliminate minor ambiguities. The introduction provides a solid overview, placing the genus Microlaimus within its taxonomic context and adequately explaining the relevance of the work in resolving existing classification uncertainties. The manuscript follows a logical structure typical for a taxonomic revision article. Figures and tables are generally well-prepared, although figure labeling could be improved by providing more detailed captions, making it easier for readers to interpret them independently. All necessary raw data and supplementary files have been transparently shared, aiding reproducibility and validation of the presented work.

Experimental design

The study design is robust and systematically addresses the primary objective of clarifying existing taxonomic confusion within the genus Microlaimus. Authors conducted a comprehensive analysis involving detailed morphological comparisons of multiple species within the genus, drawing from both existing literature and new morphological assessments. The methods chosen for the taxonomic evaluation are appropriate, clearly described, and methodically implemented. The approach involves careful examination and comparison of diagnostic morphological characteristics across species, highlighting key traits that inform taxonomic revisions. Additionally, the integration of previous literature and established databases ensures that the current investigation aligns well with prevailing standards and practices in the field. However, further elaboration on criteria used for including or excluding certain taxa from the analysis would strengthen methodological clarity.

Validity of the findings

The validity of the study's findings is underpinned by comprehensive morphological evidence and rigorous comparisons across a broad set of specimens. The authors successfully justify their taxonomic decisions, notably in reassigning species to different genera, based on clearly outlined morphological criteria and critical analysis of existing literature. Conclusions logically follow from the presented evidence, effectively resolving some long-standing ambiguities regarding species identification and classification. There is room for minor improvement in clearly separating interpretations from established facts within the discussion to reinforce scientific rigor.

Additional comments

To improve the manuscript, I suggest the authors consider enhancing the figure captions by providing greater detail to facilitate independent interpretation of visual data. Simplifying certain complex morphological descriptions could make the manuscript more accessible to a broader scientific audience. Additionally, incorporating a brief section discussing ecological or practical implications of their taxonomic clarifications would significantly increase the relevance of this review to applied research areas. Finally, providing a concise summary of recommendations for future research based on these findings could further strengthen the manuscript’s impact.

Reviewer 3 ·

Basic reporting

The manuscript presents an attempt to systemize the genus Microlaimus and construct an illustrated guide for species. This is a difficult but actual task: the genus Microlaimus harbours almost a hundred species; most of them are minute in size and differ from one another with fine morphometric characters. On the other hand, Microlaimus species are very common and may occur in almost each sediment sample taken from sea shallows. I strongly support the idea to create such a guide.
The concept of clustering species in groups and subgroups together with construction of sets of standardized images is highly promising. I hope, the guide should work.
As for essential remarks, I have only one. (line 128) “In cases where the Amph ant/hd ratio of the species presented variation that could place it in two distinct groups, the group in which the species was inserted was chosen …”. Of course, it is difficult to distinguish distinct groups on morphometric separated by gaps in such a large taxon as Microlaimus whose species differ from each other mainly by morphometric criteria. I guess, some species will inevitably tend to occupy intermediate positions. As a potential user of the illustrated guide, I believe, the best way would be to place such intermediate species in both adjacent groups.

Particular remarks
line 43. should be Kovalyev instead Kolvayev
line 134 and throughout the text. It should be more correct to replace “esophagus” and “esophageal” by “pharynx” and “pharyngeal”, according to recommendation of A. Coomans.
line 156 and throughout the text. Citing Vietnamese names presents some problem. Nguyen is a family name. I guess, since 50% Vietnamese people (if not more) bear the family name Nguyen, it would be rationally to avoid confusion by citing those names in full, e.g. Nguyen Vu Thanh.
line 185. Microlaimus dentadus should be corrected to Microlaimus dentatus
line 435. “bear-beitet” should be corrected to “bearbeitet”

Experimental design

no comment

Validity of the findings

no comment

·

Basic reporting

The manuscript "Review of the genus Microlaimus de Man, 1880 with an illustrated guide for species identification" by Esteves and co-authors proposes an illustrated guide for intraspecific identification of the genus Microlaimus. Microlaimus is a very abundant and cosmopolitan nematode genus that can be found in shallow and deep-sea ecosystems, predominantly marine but that it can be also found in brackish water and in soil. I agree with the authors that this genus, and more generally, the Microlaimidae family, is particularly complex for taxonomical ID. This ms can be really helpful for taxonomists and meiobenthologists for making correct and more precise ID of marine nematodes. The taxonomical work is outstanding and the illustrated guide very useful even for not-expert taxonomists. For all these reasons I strongly support the publication of this ms. However when consider the scope of PEERJ, I feel that the ms is not enough broad to fulfil with the scope of the journal. I suggest the authors to include a review of the genus more from an ecological-functional point of view (where we can find it?, associated with which kind of environment?, what we know about life cycles/physiology/feeding mode?, any information about adaptations to specific conditions? what about its response to human stresses?). I think that with a broad review of the importance of this genus, the taxonomical work can have a further valorisation and the ms can fit better with the scope of the journal. It would be very interesting to also have an illustrated guide/key for the family, highlighting clearly the structures responsible of differences between the different Microlaimidae genera (e.g. peribuccal bulb in Bolbolaimus). Concerning figures, some traits in drawing are too thick (e.g. Fig 2; the spicule of M. Capillaris), please make the same thickness in order to be visually coherent.

I really believe that taxonomy is necessary and usually neglected, and I recommend to Editors to support this ms that does an effort to make taxonomy more affordable. On another hand, it's important that taxonomists do the effort to make this discipline more cross-disciplinary and more attractive for a larger audience. I think that this possible and I strongly encourage to authors to do the proposed changes. For conclude, I strongly recommend this ms for publication after minor revision for fit in the broad and cross-disciplinary scope of the journal.

Experimental design

no comment

Validity of the findings

no comment

Additional comments

no comment

---

## Round 0.2 · Minor Revisions

Dear Dr. Esteves,

Please take into account the reviewer's fundamental comments and improve the manuscript. I hope that the new version of this article will be approved by the reviewer.

Reviewer 1 ·

Basic reporting

The scientific work is of interdisciplinary interest and falls within the scope of the journal. The topic is relevant for specialists in the field of zoology, parasitology and ecology. Therefore, the issues considered by the authors of the article require attention. The Introduction section describes the topic of the study quite accurately, as well as the issues that still remain unclear in this area and ways to solve this problem. The authors took into account the comments and added information about the habitat, distribution of nematodes of the genus Microlaimus de Man, 1880, and their role in the ecosystem to this section.

Experimental design

The sources in the scientific work are correctly cited, the review is logically organized into related sections and subsections. The authors of the work took into account the comments and added updated information to the list of sources and to the title of the figures, and also changed the order of the subsections.

Validity of the findings

The article has a well developed and supported argument that meets the goals set out in the Introduction. The conclusion identifies unresolved issues and future directions. The authors have taken into account all comments and made revisions to the conclusions of the research manuscript.

Additional comments

The authors took into account all the comments and wishes and made their own corrections to the manuscript. Therefore, the article should be accepted as is.

Reviewer 2 ·

Basic reporting

In the introduction, the authors provide a comprehensive overview of the taxonomic and practical identification challenges associated with the nematode genus Microlaimus de Man, 1880. This line of research is highly relevant to the study of marine biodiversity in the world's oceans. The article is written in professional English and employs modern scientific terminology. The authors have developed a tool for intraspecific identification within the genus Microlaimus. The illustrated guide presented in the article includes valid species and organizes them into groups based on morphological traits. The selection of these traits was guided by their ease of visualization and practical utility. Notably, the authors identified the morphological characters that are most informative for intraspecific differentiation. The article is of broad interdisciplinary interest and aligns well with the journal's scope. The study refines the taxonomy of the genus Microlaimus, which is valuable for both systematics and the advancement of marine biology. The practical identification tool proposed by the authors is designed for application in ecological research. This study contributes to the development of fundamental taxonomy by addressing methodological issues in species differentiation through the implementation of a standardized and illustrative approach. However, it should be noted that the sentences in lines 42 and 61 are repetitive in meaning.

Experimental design

The authors conducted a comprehensive taxonomic review of the genus Microlaimus to support their study. This process utilised modern databases and a substantial number of literature sources. The authors verified the validity of the described species and developed an illustrated system for their practical identification. The species were classified based on visually observable morphological traits commonly used in taxonomic descriptions. A reanalysis was also performed for species previously assigned to genera morphologically similar to Microlaimus, allowing the authors to reassess the accuracy of their taxonomic placement. The methodology is presented and well-reasoned, supporting the results' scientific robustness. The authors justify the sources used for the taxonomic revision, encompassing both databases and published literature. Criteria for grouping species based on observable morphological features are outlined. The methodology deliberately avoids excessive morphological jargon, making the results more accessible to a broad audience of researchers and practitioners. Nevertheless, it is worth noting that, for greater usability, the taxonomic key could be supplemented with a glossary of terms, and the illustrations could include labels indicating the key morphological characters. While this information may be unnecessary for specialists in nematology, it could benefit researchers in related fields who are not taxonomists. This recommendation is not mandatory but may enhance the utility of the tool.

Validity of the findings

The article provides a significant update on a taxonomically complex genus by introducing an innovative illustrated identification tool that enhances accessibility and standardisation at the species level—this represents the study's primary impact and scientific contribution. This method greatly assists non-specialists in identifying nematodes and strengthens biodiversity research in ecological monitoring. The study reevaluates previously defined species using updated criteria. The authors effectively summarise the results of the taxonomic revision and emphasise the practical importance of the illustrated identification guide. The article primarily focuses on how the proposed classification and morphological grouping can promote more consistent and user-friendly species identification within the genus Microlaimus. The conclusions are well-supported by the taxonomic and morphological analyses conducted, reflecting the study's scope. The article presents a coherent argument that aligns with the initial objectives. The authors articulate the need for improved tools to facilitate species-level identification in Microlaimus, addressing issues related to taxonomic uncertainty and the accessibility of diagnostic features. The article's main body systematically addresses this gap by reviewing known species, reassessing their classification, and organising them into logical groups based on observable traits. The creation of the illustrated guide directly fulfils the goal of enhancing taxonomic clarity and usability.

Additional comments

Minor shortcomings were identified earlier, and once these are addressed, the article will be ready for publication.

---

## Round 0.3 · accepted · Accept

Dear Dr. Esteves, I congratulate you on the acceptance of this article for publication and hope that you will continue to submit such high-quality articles to our journal in the future.

Reviewer 2 ·

Basic reporting

All recommendations have been taken into account and the necessary corrections have been made. I recommend the article for publication.

Experimental design

All recommendations have been taken into account and the necessary corrections have been made. I recommend the article for publication.

Validity of the findings

All recommendations have been taken into account and the necessary corrections have been made. I recommend the article for publication.

Additional comments

All recommendations have been taken into account and the necessary corrections have been made. I recommend the article for publication.